# Detecting heritable phenotypes without a model using fast permutation testing for heritability and set-tests

Regev Schweiger[1], Eyal Fisher[2], Omer Weissbrod[3], Elior Rahmani[1], Martina Müller-Nurasyid [4,5,6], Sonja Kunze[7,8], Christian Gieger[7,8], Melanie Waldenberger[6,7,8], Saharon Rosset[2] & Eran Halperin [9,10]

Testing for association between a set of genetic markers and a phenotype is a fundamental task in genetic studies. Standard approaches for heritability and set testing strongly rely on parametric models that make specific assumptions regarding phenotypic variability. Here, we show that resulting *p*-values may be inflated by up to 15 orders of magnitude, in a heritability study of methylation measurements, and in a heritability and expression quantitative trait loci analysis of gene expression profiles. We propose FEATHER, a method for fast permutation-based testing of marker sets and of heritability, which properly controls for false-positive results. FEATHER eliminated 47% of methylation sites found to be heritable by the parametric test, suggesting a substantial inflation of false-positive findings by alternative methods. Our approach can rapidly identify heritable phenotypes out of millions of phenotypes acquired via high-throughput technologies, does not suffer from model misspecification and is highly efficient.

---

[1] Blavatnik School of Computer Science, Tel Aviv University, Tel Aviv 6997801, Israel. [2] School of Mathematical Sciences, Department of Statistics, Tel Aviv University, Tel Aviv 69978, Israel. [3] Department of Epidemiology, Harvard T.H. Chan School of Public Health, Boston 02115 MA, USA. [4] Institute of Genetic Epidemiology, Helmholtz Zentrum München—German Research Center for Environmental Health, Neuherberg 85764, Germany. [5] Department of Medicine I, Ludwig-Maximilians-Universität, Munich 80539, Germany. [6] DZHK (German Centre for Cardiovascular Research), partner site Munich Heart Alliance, Munich 80636, Germany. [7] Institute of Epidemiology II, Helmholtz Zentrum München - German Research Center for Environmental Health, 85764 Neuherberg, Germany. [8] Research Unit of Molecular Epidemiology, Helmholtz Zentrum München—German Research Center for Environmental Health, 85764 Neuherberg, Germany. [9] Los Angeles, University of California Los Angeles, Los Angeles 90095 CA, USA. [10] Department of Anesthesiology and Perioperative Medicine, University of California, Los Angeles 90095 CA, USA. Correspondence and requests for materials should be addressed to R.S. (email: schweiger@post.tau.ac.il)

One of the most fundamental problems in genetics is testing whether a particular phenotype is associated with a set of markers. For instance, it is often desired to understand whether a specific phenotype is heritable. The heritability of a phenotype is defined as the proportion of the variance explained by a genetic component. In this example, the set of single-nucleotide polymorphisms (SNPs) that are tested for association with the phenotype includes the entire set of SNPs of the genome; when local heritability is tested, then the set of SNPs includes the SNPs in particular regions such as chromosomes or with a particular functional annotation. Furthermore, these variants are often tested against a large number of phenotypes, such as expression profiles of genes[1–4], methlyation levels[5–8], or neuroimaging measurements[9,10]. A similar notion to heritability has been tested in other fields as well, beyond genetics. For example, in metagenomics, it is a common practice to test for an association between a phenotype and the relative abundance vector obtained from either shotgun sequencing or targeted 16S sequencing[11]. Heritability is commonly studied using the linear mixed model (LMM); this model is a linear model that implicitly assume a very small effect for each of the SNPs[12].

Naive association testing of single markers may be extremely underpowered, even in large datasets. The common technique to address this issue is set testing, which groups together markers and applies a joint test to them[13–16]. Set testing is especially important when analyzing data with rare variants. Rare variants are becoming widely available, with sequencing costs rapidly decreasing, resulting in many whole-genome sequencing studies. Rare variants are particularly important since a large part of human genetic variation can be explained by these variants.

The success of the LMM depends on the degree to which it fits the data. For example, the LMM assumes the phenotype follows a normal distribution. However, phenotypes which are discrete, multimodal, bounded, truncated, or in general whose residuals, after adjusting for covariates, do not exhibit normality, might not be suitable for use with the LMM. The same argument holds for generalized LMMs (GLMMs), which replace the normality assumption with other parametric distributions. To mitigate such issues, one can attempt to pre-process the phenotypic values to make them as Gaussian as possible (see, e.g.[17]). However, there is no guarantee that a sufficiently good transformation exists, and the dependency on the parametric model may not be robust to other types of model deviations.

Our goal is to develop a practical and globally applicable test for existence of heritability, which will apply to the test statistic calculated via the LMM mechanism, whether or not the LMM assumptions actually fit the data well. Permutation testing is a non-parametric, assumption-free method for testing the null hypothesis of sample exchangeability[18]. Such exchangeability holds, for example, for any model under which non-heritable phenotypes are independent and identically distributed across all individuals, such as the LMM with a constant covariate and zero heritability. It also holds approximately with general covariates, under many realistic settings. In such permutation testing, we repeatedly permute the labels corresponding to each individual in the phenotype (along with additional covariates), re-estimate the heritability of each shuffled dataset, and compute the proportion of permutations for which we got a higher heritability value than the original estimate. This offers an intuitive notion of significance, which has little dependency on the underlying distribution of the phenotype.

Unfortunately, permutation tests are computationally demanding, requiring the calculation of the test statistic for each permuted dataset. As a rule of thumb, accurate estimation of $p$-values of $1/M$ requires $100M$ permutations[19]. Contemporary QTL studies[3] often carry out hundreds of thousands (or more) of tests, calling for $p$-values smaller than, e.g., 0.05/100000 to establish significance after multiple testing correction. Thus, permutation-based testing of such studies will require on the order of $10^9$ permutations, a formidable task.

Previous works suggest permutation procedures as a way to calibrate the distribution of a statistic of choice, in the context of LMMs. For example, permutations are used to calibrate the likelihood ratio (LR) test statistic, assuming its distribution comes from a limited family of distributions[16] or without such assumptions[20,21]. This approach was extended to be used in association with multiple traits[22]. A less restrictive permutation scheme was used in[23] to test whether the distribution of survival endpoints varies among centers in an acute myeloid leukemia multicentre study, using the LMM for analysis. More generally, permutation and other nonparameteric bootstrapping schemes have been used to calibrate various statistics suggested for testing for a nonzero variance components in GLMMs. In ref.[24], an alternative statistic based on the score test is suggested, with a corresponding non-permutation-based parametric bootstrap test. Other works follow a similar permutation procedure to the one presented in this work, but apply it to a newly suggested T-based statistic[20] or to particular quadratic forms of the phenotype[25]. In[9], one of the considered permutation tests is similar to ours, but they do not address computational issues, which limits their ability to detect small $p$-values.

In this paper, we study the behavior of the permutation test compared to the $p$-values calculated by assuming the parametric LMM. We analyze methylation measurement profiles from the longitudinal KORA study (Cooperative health research in the Region of Augsburg), and gene expression profiles from the GTEx project. We show that large discrepancies exist between the two tests. In particular, $p$-values from the parametric test are often much smaller than those obtained from the permutation test. We show that this likely stems from model mis-specification of the LMM, which suggests a large majority of methylation sites or expression profiles with seemingly significant heritability are in fact false positives, motivating the use of permutation tests.

We then propose a fast method for permutation testing for heritability and for set testing. To do so, we address two issues. First, for each permutation, we speed up evaluation by using the derivative of the likelihood of the permuted phenotype instead of a full estimation step. Second, we use an efficient $p$-value evaluation procedure[26] based on the Stochastic Approximation Markov Chain Monte Carlo (SAMC) algorithm[27,28], which allows us to estimate the significance of a heritability estimate with a fraction of the number of permutations required by the naive method. We apply our approach to the KORA dataset, to achieve a speed up of up to eight orders of magnitude in $p$-value calculation.

## Results

For a phenotype $\mathbf{y}$, the LMM assumes $\mathbf{y} = \mathbf{X}\boldsymbol{\beta} + \mathbf{g} + \mathbf{e}$, where $\mathbf{X}\boldsymbol{\beta}$ is the contribution of covariates, $\mathbf{g} \sim \mathcal{N}(\mathbf{0}, \sigma_g^2 \mathbf{K})$ is the genetic component of the trait, and $\mathbf{e} \sim \mathcal{N}(\mathbf{0}, \sigma_e^2 \mathbf{I})$ is the environmental component. $\mathbf{K}$ is a kinship matrix capturing the genetic relatedness between individuals in the sample, which can be constructed in various ways, using genotypes or known familial relations. Heritability is then defined as $h^2 = \sigma_g^2/(\sigma_g^2 + \sigma_e^2)$. Similarly, the heritability estimate is $\hat{h}^2 = \hat{\sigma}_g^2/(\hat{\sigma}_g^2 + \hat{\sigma}_e^2)$, where $\hat{\sigma}_g^2$ and $\hat{\sigma}_e^2$ are estimates for $\sigma_g^2$ and $\sigma_e^2$, respectively. These estimates are calculated using restricted maximum likelihood (REML) estimation (see Methods).

Under the LMM, the common technique for parametric $p$-value calculation (e.g., in the popular GCTA software package[29]) is to calculate the generalized likelihood ratio test statistic, and to

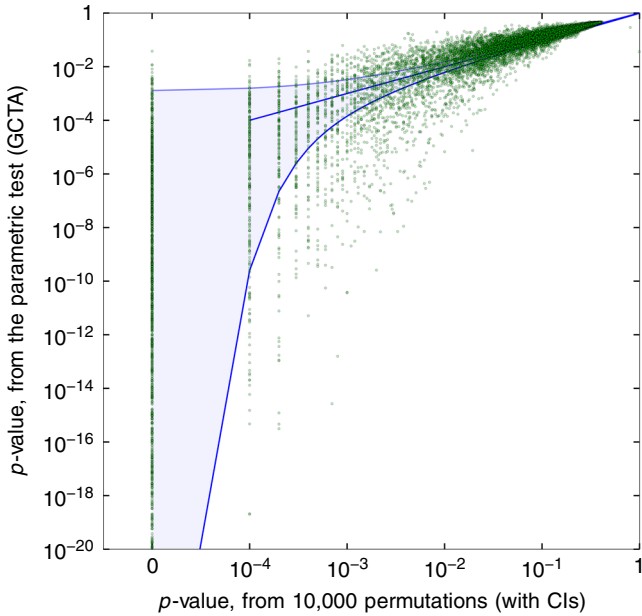

**Fig. 1** Discrepancy in *p*-values in a methylation study. *p*-values from 10,000 permutations, compared to GCTA *p*-values assuming asymptotics (in log scale). Evaluated on 431,366 methylation sites on all autosomal chromosomes, from the KORA dataset, with 1799 individuals, and with sex, age, and smoking status as covariates. Sites with $\hat{h}^2 = 0$ or with a parametric $p < 10^{-20}$ omitted for clarity of presentation, with 99.995% confidence intervals (CIs) shown. Parametric *p*-values are often smaller than the exact *p*-values obtained by the permutation test, frequently by several orders of magnitude, resulting in many false positives

assume it is distributed as a 50:50 mixture of zero and the chi-square distribution (see Methods). Other distributions are sometimes used[16], as we discuss below. An alternative test, usually used in the context of set testing, is the score test for $\sigma_g^2 = 0$, used most predominantly as implemented by the Sequential Kernel Association Test (SKAT)[30]. The distribution of the SKAT test statistic is assumed to be a certain weighted mixture of chi-square distributions.

To avoid model mis-specification or relying on an asymptotic distribution of the test statistic, we consider the permutation test. In the context of heritability, it consists of permuting the phenotype, estimating the heritability for each permuted phenotype, and counting the proportion of permutations for which the estimate obtained was higher than that of the original phenotype (see Methods). In practice, we do not enumerate over the entire set of permutations, but rather sample random permutations (e.g., $N = 1,000,000$). This gives us an estimate of the *p*-value of the test under the model, for which we can construct accurate confidence intervals (CIs). One appealing property of the permutation test is that it is an exact test under the assumption that the true model for the data is invariant to relabeling of individuals, which holds specifically if the trait is non-heritable, and holds approximately when non-constant covariates are used, in a wide range of settings. For example, if we decide that a *p*-value of < 0.001 is our threshold for proclaiming a phenotype as heritable, then we will falsely label non-heritable phenotypes as heritable about 0.1% of the time.

**Large discrepancies in *p*-values in a methylation study.** We compared the *p*-values of the permutation test to the *p*-values calculated by assuming the parametric LMM. We analyzed methylation measurement profiles from the longitudinal KORA study (Cooperative health research in the Region of Augsburg),

which consists of subjects from the general population living in the region of Augsburg, southern Germany. In this dataset, both whole-blood methylation levels and genotypes are available for 1799 individuals (see Methods). The phenotype in this study is the proportion of methylated samples at a specific site, averaged across cells. These proportions are often empirically bimodal for a given site, and their values are bounded between 0 and 1, and thus it is not clear that an assumption of normality or near-normality is suitable here.

Several works have studied heritable DNA methylation effects and the role of such epigenetic variants in disease and genetic regulation (see, e.g., a recent review[5] and references therein). For example, heritable methylation sites were previously shown[6] to be enriched for open chromatin regions and binding sites of regulators of transcription and chromatin architecture, and to be proximal to genes enriched in several known pathways, suggesting a potential regulatory mechanism through which genetic variation can affect phenotype.

We calculated the LMM heritability estimates for 431,366 methylation sites, and calculated their *p*-values using two methods: The parametric test (using the generalized likelihood ratio test, e.g., using GCTA) and using a permutation test with 10,000 permutations, using only an intercept covariate (see Methods). We observed that parametric *p*-values are often considerably smaller than the exact *p*-values obtained by the permutation test, frequently by several orders of magnitude, and thus they may incur false-positive findings (Supplementary Figure 1)). We re-ran this analysis using age, sex, and smoking status as covariates, which are commonly used in methylation studies as known confounders. The results (Fig. 1) show large discrepancies, showing that it is not a result of lack or addition of covariates in the analysis.

Applying a Bonferroni threshold of $0.05 \cdot 1/431,366 \approx 10^{-7}$, we further took the 3489 most significant sites which passed this threshold accordingly to GCTA. We then estimated their *p*-value using 10,000,000 permutations (Supplementary Figure 2). This increased accuracy reveals that for many sites, GCTA would proclaim a site as very significant, while permutation testing does not indicate significance. For example, 395 sites have a permutation *p*-value of $p > 10^{-4}$, but a parametric *p*-value of $p < 10^{-10}$. A further inspection of phenotypes displaying such large discrepancies discovered that it is often the result of the phenotype taking disparate values for individuals who are relatively genetically distant from the rest of the sample.

Finally, to check for opposite discrepancies, we took the 6686 sites with permutation *p*-value of 0/10,000, and calculated their permutation *p*-value using the method we present below. For 81 sites, the permutation *p*-value as estimated was significant while the parametric was not, indicating that parametric methods can suffer not only from false-positive results, but also from lack of power. For most of these sites (62/81 sites, 76%), the discrepancy was less than 1 order of magnitude smaller, which can be accounted for by noise in *p*-value estimation. The remaining 19 sites showed discrepancies below 2 orders of magnitude, and 14 of them exhibit a tri-modal behavior. The results are summarized in Table 1.

**Large discrepancies in *p*-values in a gene expression study.** In order to show the generality of the discrepancy phenomenon, we analyzed gene expression data in the GTEx dataset (see Methods). We first performed a cis-eQTL study, where for each of 22,171 genes, we created a kinship matrix from the SNPs located within 500 kbps from the gene transcription start site (total window size of 1 Mbps). For each gene, we used both the parametric and the permutation tests to test for association between the SNPs in the

**Table 1 Summary of *p*-value discrepancies**

|  | Non-significant (GCTA) | Significant (GCTA) |
| --- | --- | --- |
| Non-significant (Perm.) | 427,796 | 1667 |
| Significant (Perm.) | 81 | 1822 |

The *p*-values for heritability of 431,366 methylation sites, as calculated by either GCTA (using the parametric approach) or the permutation test, are shown. Sites are considered significant if their *p*-values are below a Bonferroni threshold of $10^{-7}$

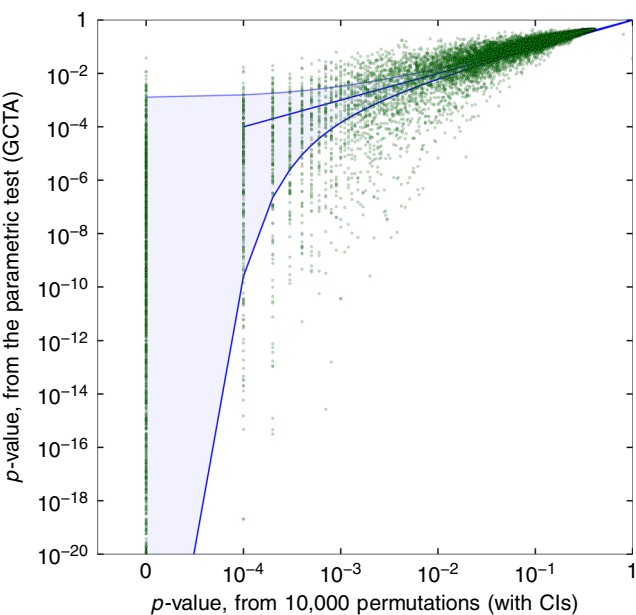

**Fig. 2** Discrepancy in *p*-values in a cis-eQTL study. *p*-values from 10,000 permutations, compared to GCTA *p*-values assuming asymptotics (in log scale). Evaluated on 22,171 gene expression profiles in whole-blood samples, from the GTEx dataset, with 338 individuals. Sites with $\hat{h}^2 = 0$ (8604 profiles) omitted for clarity of presentation, with 99.995% CIs. Parametric *p*-values are often smaller than the exact *p*-values obtained by the permutation test, frequently by several orders of magnitude, resulting in many false positives

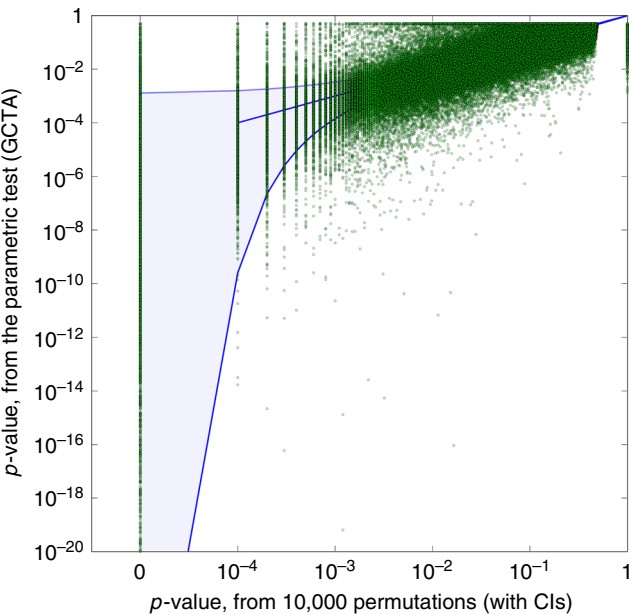

**Fig. 3** Discrepancy in *p*-values, with quantile normalization. *p*-values after quantile normalization, from 10,000 permutations, compared to GCTA *p*-values assuming asymptotics (in log scale). Parametric *p*-values show large discrepancies compared to the exact *p*-values obtained by the permutation test, frequently by several orders of magnitude, resulting in many false positives and negatives

window and the gene expression profile, as measured in whole-blood samples. We used the same covariates used in the original GTEx study—namely, the first three genotyping principal components (PCs), the first 15 expression PEER factors (Probabilistic Estimation of Expression Residuals)[31], and sex. Gene expression profiles were quantile-normalized before analysis (see[32]). As evident in Fig. 2, despite inclusion of relevant covariates and careful preprocessing, there still remain significant discrepancies that could be a source of false positives. Indeed, in the original GTEx study, one required criterion for detection of eQTL-containing genes is a permutation *p*-value obtained from 10,000 permutations[32]. This exemplifies the need for a fast and accurate permutation testing procedure.

In addition, we performed a heritability study over the same data, where now the set of tested markers includes the entire genome. We still observed *p*-value discrepancies, when permutation *p*-values did not detect significantly heritable expression profiles, while parametric *p*-values did. Further analysis showed that all profiles with significant parametric *p*-values obtained a heritability estimate of $\hat{h}^2 = 1$ (Supplementary Figure 3). In such a case, the assumed LRT statistic distribution (see Methods) does not handle the maximal boundary estimate correctly. In studies of small sample sizes, such boundary estimates are likely to occur[33],

so such discrepancies are expected. This presents another scenario where parametric *p*-values are not calibrated.

**Reasons for *p*-value discrepancy**. We proceeded to analyze possible underlying reasons for *p*-value discrepancies. The analysis is given in Supplementary Note 1, and is summarized here. First, we showed that the permutation test is equally powerful to the parametric test under the LMM (Supplementary Figure 4), so that power differences do not explain the discrepancy. Second, we applied quantile normalization (QN) as a preprocessing step before calculating *p*-values, finding that it did not eliminate the discrepancies, and introduced a potential power loss (Fig. 3). Third, we considered an extended family of distribution as an alternative to the assumed distribution of the LRT statistic[16], but it failed to substantially alter the results. Fourth, we verified that sites with *p*-value discrepancies are not limited to those with a multimodal behavior. To this end, we excluded from the analysis both sites whose probes are known to contain SNPs (and are thus expected to be multimodal), as well as sites empirically showing a multimodal behavior. However, *p*-value discrepancies remained. Fifth, we examined whether this discrepancy exists when using the score test instead of the LR test. We note that using the SKAT method[30] to calculate *p*-values, we found that it generated significantly deflated *p*-values throughout the dataset, indicating that the statistic distribution is not calibrated[34,35]. Using RL-SKAT[35]

**Table 2 Benchmarks**

| Algorithm | GCTA | pylmm | FaST-LMM | FEATHER (Deriv. Based) | FEATHER (SAMC) |
|---|---|---|---|---|---|
| No. of perm. used $p = 10^{-4}$ | $10^6$ | $10^6$ | $10^6$ | $10^6$ | $10^6$ |
| Total time used $p = 10^{-4}$ | >8 months | ~1.8 days | ~13 h | ~23 min | ~16 min |
| No. of perm. used $p = 10^{-7}$ | $10^9$ | $10^9$ | $10^9$ | $10^9$ | $10^6$ |
| Total time used $p = 10^{-7}$ | | >4 years | ~1.4 years | ~16 days | ~16 min |
| No. of perm. used $p = 10^{-9}$ | $10^{11}$ | $10^{11}$ | $10^{11}$ | $10^{11}$ | $10^6$ |
| Total time used $p = 10^{-9}$ | | | | >4 years | ~16 min |

The running time of permutation tests, as calculated by GCTA[29], pylmm[37] and FaST-LMM[38] (performing full estimation for each permutation), vs. the derivative-based approach, with and without SAMC, on a single site from the KORA dataset, with 1799 individuals. Benchmarks are shown for the cases where the true $p$-value is $10^{-4}$, $10^{-7}$, or $10^{-9}$. The number of permutations required for the non-SAMC methods is assumed to be $p/100$ (see Methods). For these $p$-values, SAMC converged after $10^6$ permutations (see Results). Per permutation, the derivative-based approaches have a speed up of two orders of magnitude compared to the best full estimation permutation test program. The usage of SAMC for small $p$-values introduces an additional speed up of up to 6 orders of magnitude. Expected computation times for several sites are multiplied by the number of examined sites

to calculate calibrated $p$-values, we observed the same discrepancies (Supplementary Figure 5). Finally, we performed a simulation study, where non-heritable phenotypes were generated from a non-normal, heavy-tailed marginal distribution per entry. These phenotypes showed parametric $p$-value miscalibration, while permutation $p$-values remained calibrated.

We conclude that model mis-specification is the probable reason for such large discrepancies, which further motivates the use of the permutation test. We find that if the LMM is a suitable model for the data at hand, then both the parametric and the permutation tests have similar power. In the case where the LMM is not suitable, the parametric test breaks while the permutation test remains calibrated under much weaker assumptions. Therefore, from a statistical perspective, the permutation test is superior in the context of testing for heritability as defined by the LMM.

**Speeding up the evaluation step per permutation**. Running time is a major obstacle to performing permutation tests. First, for each permutation, finding the heritability estimate of the permuted phenotype is a computationally intensive task. Second, in order to accurately estimate small $p$-values, we would need to draw many permutations. In our proposed method, FEATHER (Fast pErmutAtion Testing HERitability), we address these two considerations in turn, beginning with the task of speeding up the evaluation performed for each random permutation.

For each permuted phenotype, the naive permutation test estimates its heritability and compares it to the heritability of the unpermuted phenotype, denoted $H^2$. However, we are not in fact interested in the estimated heritability value of the permuted phenotype, but rather only if it is smaller or larger than that of the unpermuted phenotype. Recall that REML obtains the heritability estimate which maximizes the (restricted) likelihood of the permuted phenotype as a function of the suggested heritability value. Consider the derivative of the likelihood function at the point $H^2$. Assuming the likelihood function is well behaved, this derivative points us to the direction of the maximum: If the derivative is positive, then the maximum is obtained at a value larger than $H^2$, and conversely if it is negative (see Supplementary Figure 6 for an illustration). Therefore, a faster approach is to simply examine the derivative of the likelihood function, rather than trying to find its maximum. We validated the assumption that the likelihood function is well behaved in practice with extensive simulations of real and permuted phenotypes (Supplementary Note 2).

Additionally, in our previous work[33] it was shown that given the eigendecomposition of the kinship matrix, the derivative of the likelihood function can be calculated in $O(n^2)$ time. Moreover, the core of the computation is a single matrix-by-vector product, an operation enjoying an efficient implementation in

existing software and hardware[36], and thus a small constant factor (see Methods). The savings in computation complexity depend on the heritability estimation algorithm used in the naive approach. For example, when using the AI or the EM algorithms, as in GCTA[29], the computational complexity is $O(n^3)$, which gives our approach a speed up factor of $O(n)$. With approaches that utilize the eigendecomposition, such as pylmm[37] or FaST-LMM[38], the asymptotic complexity is the same; However, we still get a significant empirical speed up as a result of avoiding many evaluations of the likelihood functions. In practice, on the KORA dataset, we observed a speed up of four orders of magnitudes per permutation of the derivative-based approach compared to full estimation using the widely used GCTA tool, and an order of magnitude improvement compared to FaST-LMM (Table 2).

**Reducing the number of sampled permutations with SAMC**. A major computational hurdle of permutation testing is the potentially large number of random permutations that need to be used in order to estimate small $p$-values accurately. To cope with this computational burden, we use an efficient $p$-value evaluation procedure based on the Stochastic Approximation Markov Chain Monte Carlo (SAMC) algorithm[26,27]. We give an overview of the method and its properties here. For the full description, see Supplementary Note 4.

In the context of heritability testing, we utilize SAMC as follows. Given an estimate $H^2$, we want to calculate its $p$-value, i.e., the probability a randomly permuted phenotype obtains a higher heritability estimate. We divide the interval $[0,1]$ to $D + 1$ intervals, where the interval $[0,H^2]$ is divided into $D$ equally sized intervals, and $[H^2,1]$ is an additional interval. This induces a partitioning of the permutation space to $D + 1$ subsets. Each subset is the set of permutations of the phenotype for which the estimated heritability value falls in the corresponding interval. Then, the $p$-value is exactly the size of the subset corresponding to $[H^2,1]$, divided by $n!$, the number of permutations of size $n$.

The SAMC algorithm estimates the size (i.e., probability) of each subset in the partition. Starting with an arbitrary initial permutation, each SAMC iteration consists of two steps: (1) Given the current random permutation, sample a new random permutation according to a certain target distribution, using the Metropolis-Hastings (MH) sampling algorithm; (2) Given the subset in which the new permutation falls, update the partition probability estimates, and the target distribution of step 1. The update rule for subset probability estimates follows the stochastic approximation algorithm[39], which ensures that the estimates can be improved continuously as the simulation goes on. Importantly, the number of random permutations required for convergence is much smaller than the estimated $p$-value.

Additionally, in order to determine in which subset a permutation falls, we need not calculate the heritability estimate

of the respective permuted phenotype. Instead, we can check the derivatives at the endpoints of the interval. If the derivative is positive at the left endpoint and negative at the right endpoint, then we know a maximum exists within that interval. Using the derivative allows us to avoid the heritability estimation step, as before. When the algorithm converges, its estimate for the last subset will be our estimate of the $p$-value.

**Analysis of the performance of SAMC.** We implemented both the simple and SAMC derivative-based permutation testing, as an efficient, multi-threaded C++ program. The fact that only few of the values of the random permutation change between successive iterations allows SAMC to be faster than the standard permutation testing, per permutation (see Table 2).

SAMC has several parameters that need to be tuned for a successful application. The most important of which are the number of intervals, $D$; the number of iterations, $N$; and an additional parameter, $t_0$, that corresponds to the number of iterations after which the estimation will begin converging more rapidly. As described in Supplementary Note 4, we chose $t_0 = 1,000$ and $N = 1,000,000$ as suitable parameters. Here and throughout the rest of the paper, we used $D = 50$ intervals.

In Fig. 4 we show the SAMC $p$-value estimates, compared to those from a standard permutation testing with 10,000 permutations, across all methylation sites in chromosome 22. To further validate the accuracy of SAMC on smaller $p$-values, we ran SAMC on the sites which GCTA deemed significant, as described above. Again, SAMC appears to give accurate estimates, also for small $p$-values, where such accuracy is more important in practical applications, as shown in Supplementary Figure 7. Finally, we chose sites with particularly small $p$-values, and compared their parametric $p$-values to those by the permutation test with $10^9$ permutations, and with the $p$-values estimated by SAMC. In Table 3, we show 10 sites with permutation $p$-values larger than 0, which enable us to informatively examine the calibration of SAMC. The results suggest that SAMC continues to give accurate permutation $p$-value estimates, as far as it was possible for us to assess.

While SAMC is guaranteed to converge[27], there are no theoretical guarantees of the speed of convergence. In practice,

we have observed that the number of required permutations is significantly reduced, with 2-to-5 orders of magnitude. This is in line with previous applications of SAMC[26]. In summary, the derivative-based approach gives a speed up improvement of at least two orders of magnitude over full estimation approaches; additionally, for small $p$-values, the SAMC approach may improve by up to another six orders of magnitude. These improvements allow testing for heritability without the assumption of a parametric model, in a feasible time.

**Discussion**

In this work, we have discussed the merits of permutation testing for heritability, compared with parametric methods. We have presented two ways to accelerate permutation testing: First, using the derivative of the likelihood function in order to avoid finding the maximum likelihood estimator; and second, using SAMC to substantially reduce the required number of permutations. We have shown that with these modifications, the running time decreases by several orders of magnitude.

SAMC requires using a minimal number of permutations before convergence. Therefore, given a large set of phenotypes for which we wish to test for heritability, we suggest the following scheme. First, perform simple (non-SAMC) permutation testing with a small (e.g., 100) number of permutations. Filter out all sites whose permutation $p$-value was too large, e.g., for which the lower end of a one-sided binomial or Poisson confidence interval is larger that the threshold $p$-value indicating significance. Continue with this gradual filtering, increasing the number of permutations in each round. Once reaching $N$ large enough for SAMC convergence (calibrated as described above), switch to SAMC for estimating the $p$-value for the remaining sites.

One advantage of the permutation approach is that it allows using certain statistics while overlooking otherwise important methodological caveats. For example, it is known that using the REML estimator for ascertained case-control studies leads to incorrect estimates[40]. However, under the mild conditions considered here, the probability of false positives will remain calibrated, although the test may be underpowered. Indeed, any statistic that captures a correspondence between a phenotype and genetics will be suitable here.

We note that our use of SAMC is also independent of our choice of the REML estimator as the statistic. Indeed, any other statistic, for which it is possible to determine in which region of a partition the statistic of a permuted phenotype falls, can be used. Natural candidates are the score test statistic[30], or the PCGC regression[40] statistic.

One critical issue that is not covered by our current approach is the limitation to one variance component. Many applications currently use LMMs using multiple variance components, specifically by dividing the genome into regions and constructing a variance component from each region. In those cases, the contribution of each variance component is then estimated or tested for being significant. Estimating multiple variance components with REML is computationally intensive, and the derivative does not appear to lend itself to a simple analytical expression as in the single variance component case. However, as PCGC regression provides an alternative, faster estimation method, using its statistic is a particularly attractive avenue of research in the context of multiple variance components.

Typically, a preliminary step in heritability estimation is the eigendecomposition of the kinship matrix. This procedure is not very efficient, as it is cubic in the number of individuals, and may therefore be too computationally inefficient for large datasets that include many individuals. Recently, it has been suggested to use conjugate gradient methods in order to estimate heritability[41],

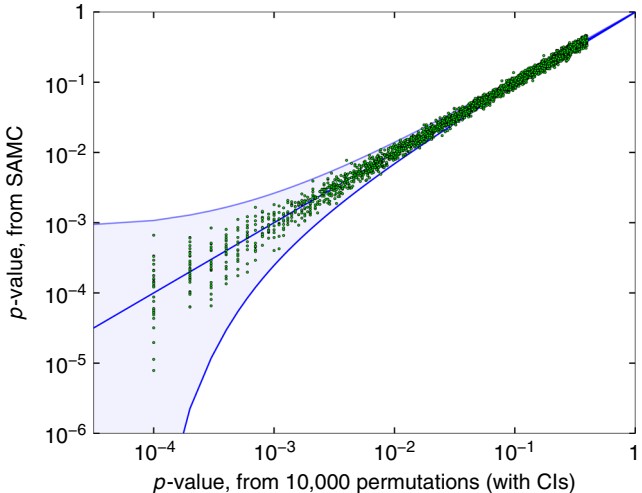

**Fig. 4** Performance of SAMC. $p$-values from 10,000 permutations, compared to SAMC $p$-values with $t_0 = 1,000$ and 1,000,000 permutations (in log scale). Evaluated on 7989 methylation sites on chromosome 22, from the KORA dataset. Sites with $\hat{h}^2 = 0$ (3,779 sites), omitted for clarity of presentation, showing a total of 4210 sites, with 99.95% CIs shown. SAMC is well calibrated

**Table 3 Performance of SAMC on extreme *p*-values**

| Site | Parametric *p*-value | Perm. *p*-value | CI | SAMC, $N = 10^5$ | SAMC, $N = 10^6$ |
|---|---|---|---|---|---|
| cg00123214 | $2.601 \cdot 10^{-08}$ | $12 \cdot 10^{-09}$ | $(6.201, 20.962) \cdot 10^{-09}$ | $6.301 \cdot 10^{-09}$ | $13.71 \cdot 10^{-09}$ |
| cg00044796 | $3.779 \cdot 10^{-13}$ | $4 \cdot 10^{-09}$ | $(1.090, 10.242) \cdot 10^{-09}$ | $6.092 \cdot 10^{-09}$ | $5.065 \cdot 10^{-09}$ |
| cg01821635 | $3.941 \cdot 10^{-18}$ | $2 \cdot 10^{-09}$ | $(0.242, 7.225) \cdot 10^{-09}$ | $4.500 \cdot 10^{-09}$ | $6.504 \cdot 10^{-09}$ |
| cg06784218 | $5.756 \cdot 10^{-21}$ | $4 \cdot 10^{-09}$ | $(1.090, 10.242) \cdot 10^{-09}$ | $2.327 \cdot 10^{-09}$ | $6.484 \cdot 10^{-09}$ |
| cg08963013 | $4.625 \cdot 10^{-26}$ | $6 \cdot 10^{-09}$ | $(2.202, 13.059) \cdot 10^{-09}$ | $3.339 \cdot 10^{-09}$ | $5.460 \cdot 10^{-09}$ |
| cg14252149 | $5.005 \cdot 10^{-34}$ | $4 \cdot 10^{-09}$ | $(1.090, 10.242) \cdot 10^{-09}$ | $3.358 \cdot 10^{-09}$ | $5.131 \cdot 10^{-09}$ |
| cg00071950 | $5.678 \cdot 10^{-36}$ | $5 \cdot 10^{-09}$ | $(1.623, 11.668) \cdot 10^{-09}$ | $8.149 \cdot 10^{-09}$ | $7.440 \cdot 10^{-09}$ |
| cg02002194 | $9.238 \cdot 10^{-42}$ | $4 \cdot 10^{-09}$ | $(1.090, 10.242) \cdot 10^{-09}$ | $6.420 \cdot 10^{-09}$ | $7.501 \cdot 10^{-09}$ |
| cg11266682 | $1.873 \cdot 10^{-46}$ | $7 \cdot 10^{-09}$ | $(2.814, 14.423) \cdot 10^{-09}$ | $12.62 \cdot 10^{-09}$ | $6.251 \cdot 10^{-09}$ |
| cg15296535 | $8.427 \cdot 10^{-51}$ | $9 \cdot 10^{-09}$ | $(4.115, 17.085) \cdot 10^{-09}$ | $6.839 \cdot 10^{-09}$ | $9.045 \cdot 10^{-09}$ |

For 10 selected sites, we show the parametric p-value, as calculated by GCTA; the permutation p-value, as calculated by $10^9$ permutations, and the CIs implied by it; and the p-values estimated by SAMC, using $N = 10^5$ or $N = 10^6$ permutations (with $t_0 = 1000$). SAMC estimates, even for $N = 10^5$, appear accurate for all sites, as they lie within the CI calculated from the permutation test

thus avoiding the cubic complexity. A natural extension of our proposed method is to derive a procedure that calculates the derivative of the restricted likelihood function using conjugate gradient methods. Such a procedure could result in a quadratic complexity, as it avoids the eigendecomposition.

One disadvantage of SAMC is that, unlike the simple permutation test, it has to be run sequentially. However, it is possible to run multiple shorter chains simultaneously, each with a less strict convergence criteria, and then to aggregate the results in order to obtain a more accurate estimate[26]. Preliminary results appear encouraging, but a more thorough study of this tradeoff remains a direction for future research, as well as other variants of SAMC[42].

Finally, the ideas presented here can be readily extended to the testing of genetic correlations, which determine if there is evidence that two phenotypes have common underlying genetic drivers (e.g., two diseases or two gene expression profiles)[43].

## Methods

For clarity of presentation, we begin by defining the heritability under the LMM. We then introduce our improved method for fast permutation testing for heritability, while reviewing relevant results from the ALBI method[33] and the SAMC algorithm.

**The linear mixed model**. We first present the standard variance components model[44]. Let $n$ be the number of individuals (or observations, in general) and let $\mathbf{y}$ be a $n \times 1$ vector of phenotype measurements for each individual. Let $\mathbf{X}$ be a $n \times p$ matrix of $p$ covariates, associated with fixed effects (possibly including an intercept vector $\mathbf{1}_n$ as a first column, as well as other covariates, such as sex, age, etc.). Let $\boldsymbol{\beta}$ be a $p \times 1$ vector of fixed effects. Let $\mathbf{Z}$ be a $n \times m$ standardized (i.e., columns have zero mean and unit variance) genotype matrix containing the $m$ SNPs we test. Finally, let $\mathbf{K}$ be a kinship matrix, which can be taken to be any symmetric positive-definite matrix that encodes similarity between individuals, using any biomarkers, e.g., a set of SNPs[12]; formally, define $\mathbf{K} = \mathbf{ZWZ^T}$, where $\mathbf{W}$ is a non-negative $m \times m$ diagonal matrix assigning a weight per SNP (e.g., $\mathbf{W}_{i,i} = 1/m$, see[30] for a discussion). Then, $\mathbf{y}$ is assumed to follow:

$$\mathbf{y} \sim \mathcal{N}\left(\mathbf{X}\boldsymbol{\beta}, \sigma_g^2 \mathbf{K} + \sigma_e^2 \mathbf{I}_n\right), \quad (1)$$

The fixed effects $\boldsymbol{\beta}$ and the coefficients $\sigma_g^2$ and $\sigma_e^2$ are the parameters of the model.

The narrow-sense heritability due to genotyped common SNPs is defined as the proportion of total variance explained by a genetic component[45]:

$$h^2 = \frac{\sigma_g^2}{\sigma_g^2 + \sigma_e^2}.$$

Defining $\sigma_p^2 = \sigma_g^2 + \sigma_e^2$, Equation (1) becomes:
$$\mathbf{y} \sim \mathcal{N}\left(\mathbf{X}\boldsymbol{\beta}, \sigma_p^2 \mathbf{V}_{h^2}\right). \text{where } \mathbf{V}_{h^2} = h^2 \mathbf{K} + (1 - h^2)\mathbf{I}_n.$$

**Estimation and testing of heritability with REML**. The most common way of estimating $h^2$ is REML estimation. REML consists of maximizing the likelihood function associated with the projection of the phenotype onto the subspace

orthogonal to that of the fixed effects of the model[46]. The logarithm of the REML function is, up to additive and multiplicative constants:

$$\ell_{\text{REML}}(\mathbf{y}; h^2, \sigma_p^2, \boldsymbol{\beta}) \propto$$
$$- (n - p)\log\sigma_p^2 - \log|\mathbf{V}_{h^2}| - \log|\mathbf{X}^T \mathbf{V}_{h^2}^{-1} \mathbf{X}| - \frac{(\mathbf{y} - \mathbf{X}\boldsymbol{\beta})^T \mathbf{V}_{h^2}^{-1}(\mathbf{y} - \mathbf{X}\boldsymbol{\beta})}{\sigma_p^2},$$

In practice, often some of the eigenvalues of $\mathbf{K}$ are zero or near-zero. This occurs, for example, when $\mathbf{K}$ is constructed from a mean-centered $\mathbf{Z}$, in which case the constant vector $\mathbf{1}$ would be an eigenvector with the eigenvalue 0. Another example is when $\mathbf{Z}$ has fewer SNPs than samples, in which case $\mathbf{K}$ will be low rank. When this is the case, the likelihood at $h^2 = 1$ may be undefined or prone to numerical instability. To avoid this, we project both the phenotype and covariates to the subspace spanned by eigenvectors corresponding to nonzero eigenvalues. Let $\mathbf{U}$ be the matrix whose columns are the eigenvectors of $\mathbf{K}$, and let $d_i$ be the eigenvalues of $\mathbf{K}$, for $i = 1, \dots, n$. If there are $z$ eigenvalues larger than a sufficiently small threshold, denote by $\mathbf{U}_z$ the matrix with the first $z$ eigenvectors. Effectively, this amounts to replacing $V_{h^2}$ in $\ell_{\text{REML}}$ with $\mathbf{U}_z^T \mathbf{V}_{h^2} \mathbf{U}_z$.

The common way to test for the statistical significance of a nonzero heritability value is using the generalized restricted likelihood ratio test statistic

$$\Lambda = \frac{\max\limits_{h^2, \sigma_p^2, \boldsymbol{\beta}} \mathcal{L}_{\text{REML}}\left(\mathbf{y}; h^2, \sigma_p^2, \boldsymbol{\beta}\right)}{\max\limits_{\sigma_p^2, \boldsymbol{\beta}} \mathcal{L}_{\text{REML}}\left(\mathbf{y}; 0, \sigma_p^2, \boldsymbol{\beta}\right)},$$

where $\ell_{\text{REML}} = \log\mathcal{L}_{\text{REML}}$. Asymptotically, we have the distribution

$$2\log\Lambda \sim 0.5 \cdot \chi_0^2 + 0.5 \cdot \chi_1^2$$

where $\chi_1^2$ is the chi-square distribution with 1 degree of freedom, and $\chi_0^2$ is the constant distribution of the constant zero[47,48].

**Permutation testing for heritability**. Monte Carlo permutation testing: The p-value of the full permutation test is calculated by enumerating over all permutations to calculate

$$p_{\text{perm}} = \frac{1}{n!}\left|\left\{\pi \in S_n, |\hat{h}^2(\pi(\mathbf{y})) \geq \hat{h}^2(\mathbf{y})\right\}\right|$$

where $\pi(\mathbf{y})$ is the application of the permutation $\pi$ on the phenotype $\mathbf{y}$, and $S_n$ is the set of all permutations of $n$ elements. This is an exact test—that is, under a null hypothesis invariant to permutations, $p_{\text{perm}}$ is distributed uniformly. However, since the number of permutations, $n!$, is huge, the common approach is to employ a Monte Carlo approximation. In detail, let $\pi_1, \dots, \pi_N$ be $N$ random permutations of $n$ elements; the p-value of the test is:

$$p_{\text{MC}} = \frac{1}{N}\left|\left\{\pi_t | \hat{h}^2(\pi_t(\mathbf{y})) \geq \hat{h}^2(\mathbf{y})\right\}\right|$$

The p-value $p_{\text{MC}}$ is an approximation of the required p-value $p_{\text{perm}}$. Moreover, since each permutation was chosen randomly and with replacement, $p_{\text{MC}}$ can be seen as the result of a binomial experiment. Therefore, we can calculate accurate confidence intervals for $p_{\text{perm}}$ given $p_{\text{MC}}$, e.g., using the Cloppe–Pearson method[49].

Covariates: If there are no covariates, or the only covariate is the constant vector (i.e., $\mathbf{X} = \mathbf{1}_n$), then the permutation test does not require taking covariates into consideration. In the general case, we apply the same permutation on each covariate vector as we do on the phenotype. We note that the permutation approach requires exchangeability of the residuals, which are not observed in general in presence of non-constant covariates in the model. However, we verified in simulations that the test remains exact or approximately exact under various settings; this is theoretically supported in recent works on permutation tests in linear regression[50–52]. See Supplementary Note 3 for an extended discussion.

**Speeding up evaluation by using the likelihood derivative**. The naive calculation above requires, for each permuted phenotype $\pi_t(\mathbf{y})$, the estimation of its heritability using REML, $\hat{h}^2(\pi_t(\mathbf{y}))$. However, instead of explicitly calculating $\hat{h}^2(\pi(\mathbf{y}))$, we are only interested whether $\hat{h}^2(\pi(\mathbf{y})) \geq H^2$, where $H^2 = \hat{h}^2(\mathbf{y})$, the heritability estimate of the unpermuted phenotype.

In[33], it is shown that when $\mathbf{X} = \mathbf{1}_n$, checking if $\hat{h}^2(\pi(\mathbf{y})) \geq H^2$ can equivalently be performed by computing $\mathbf{u} = \mathbf{U}^T\pi(\mathbf{y})$, and checking if

$$\sum_{i=1}^{n} \xi_i^{H^2} u_i^2 > 0, \qquad (2)$$

where

$$\xi_i^{H^2} = \frac{1}{H^2(d_i-1)+1}\left(\frac{d_i-1}{H^2(d_i-1)+1} - \frac{1}{n-1}\sum_{j=1}^{n-1}\frac{d_j-1}{H^2(d_j-1)+1}\right), \text{for } i = 1, ..., n-1,$$

and $\xi_n^{H^2} = 0$. The sign of the expression in Eq. (2) is equal to the sign of $\frac{\partial \ell_{\text{REML}}}{\partial h^2}(H^2)$, the derivative of $\ell_{\text{REML}}$ at the point $H^2$. Therefore, assuming the restricted likelihood function is well behaved, a positive derivative indicates that the REML heritability estimate is larger than $H^2$. Similar expressions are defined for a general $\mathbf{X}$ in[33].

Therefore, once the eigendecomposition of $\mathbf{K}$ is obtained, calculating $p_{\text{MC}}$ may be performed in a time complexity quadratic in $n$:

1. Given $\mathbf{y}$ and its heritability estimate $H^2$, calculate $\xi_i^{H^2}$ (complexity: $O(n)$).
2. Draw $\pi_1,...,\pi_N \in S_n$ (complexity: $O(nN)$).
3. For $t = 1,...,N$:

(a) Calculate $\mathbf{u}_t = \mathbf{U}^T\pi_t(\mathbf{y})$ (complexity: $O(n^2)$).
(b) Let $b_t = 1$ if $\sum_{i=1}^{n}\xi_i^{H^2}(\mathbf{u}_t)_i^2 \geq 0$ and $b_t = 0$ otherwise (complexity: $O(n)$).
4. Return $p = \frac{1}{N}\sum_{t=1}^{N} b_t$ (complexity: $O(N)$).

The total complexity is $O(n^2N)$. For a general covariate matrix $\mathbf{X}$, the only change will be in the condition checked in step (b), whose complexity is $O(np^2 + p^3)$ instead of $O(n)$, resulting in a final complexity of $O((n^2 + np^2 + p^3)N)$, as detailed in[33].

**Reducing the number of sampled permutations using SAMC**. To cope with the major computational hurdle of permutation testing, we use an efficient $p$-value evaluation procedure based on the Stochastic Approximation Markov Chain Monte Carlo (SAMC) algorithm[26,27]. A description of the SAMC algorithm and its tuning is given in Supplementary Note 4.

In summary, let the proposal distribution $q(\pi_t, \tau)$ define the probability of choosing a new permutation $\tau$, given that the current permutation is $\pi_t$. Let

$$\mathbf{e}_i = \left(0, ..., 0, \underbrace{1}_{i}, 0, ..., 0\right).$$ Let $D + 1$ be the number of intervals in the partitioning of [0,1]. For a permutation $\pi \in S_n$, let $J(\pi)$ be the index of the interval in which $\hat{h}^2(\pi(\mathbf{y}))$ falls. Let $\theta_1^{(t)}...,\theta_{D+1}^{(t)}$ be the logarithm of our current estimates of partition sizes, up to a multiplicative (in log scale, additive) constant. The algorithm is:

1. Initialize a uniform estimate, $\theta_1^{(t)} = ... = \theta_{D+1}^{(t)} = 0$.
2. Choose a random initial permutation $\pi_1$.
3. For $t = 1,...,T$ (or until convergence):

(a) Simulate a sample $\pi_{t+1}$ by a single Metropolis-Hastings update, as follows:
i. Generate $\tau$ according to the proposal distribution $q(\pi_t, \tau)$.
ii. Calculate the ratio $r = \exp\left(\theta_{J(\pi_t)}^{(t)} - \theta_{J(\tau)}^{(t)}\right) \cdot q(\tau, \pi_t)/q(\pi_t, \tau)$
iii. Accept the proposed move with a probability of min (1, $r$). If accepted, set $\pi_{t+1} = \tau$. Otherwise, set $\pi_{t+1} = \pi_t$.
(b) Update the estimates: For $i = 1,...,D + 1$, set $\theta_i^{(t+1)} = \theta_i^{(t)} + \gamma^{(t)}\left(\mathbf{e}_{J(\pi_{t+1})} - \left(\frac{1}{D+1}, ..., \frac{1}{D+1}\right)\right)$, where $\gamma^{(t)}$ is called the gain factor and is defined as $\gamma^{(t)} = t_0/\max(t_0, t)$.
4. Return $\exp(\theta_{D+1}^{(t)})/\sum_{i=1}^{D+1}\exp(\theta_i^{(t)})$.

**The KORA dataset**. The KORA project studies $n = 1799$ individuals from the general population living in the region of Augsburg, southern Germany[53]. The measured phenotype is the proportion of methylated samples at a specific site, averaged across DNA samples of an individual. We used whole-blood samples of the KORA F4 study, as described elsewhere[54]. Briefly, DNA methylation levels were collected using the Infinium HumanMethylation450K BeadChip array (Illumina). Beta Mixture Quantile (BMIQ)[55] normalization was applied to the methylation levels. Further processing was performed as in ref.[56]; briefly, genotyping was performed with the Affymetrix 6.0 SNP Array (534,174 SNP markers after quality control), with further imputation using HapMap2 as a reference panel. A total of 657,103 probes remained for the analysis. In summary, a total of 431,366 methylation site phenotypes, and 657,103 SNPs, were available for analysis. Covariates used in this study are age, sex, and smoking status.

**The GTEx dataset**. The Genotype-Tissue Expression (GTEx)[52] Project is a US National Institutes of Health (NIH) Common Fund project that aims to collect a comprehensive set of tissues from 900 deceased donors (for a total of about 20,000 samples) and to provide the scientific community with a database of genetic associations with molecular traits such as mRNA levels. We used 22,171 gene expression profiles obtained from whole-blood samples of 338 individuals, as preprocessed and using the same covariates as described in[32].

**Benchmarks**. We used GCTA version 1.26[29] and pylmm[57]. We used the C++ implementation of FaST-LMM, FastLmmC v2.07.20140723[38] and calculated the kinship matrix and its eigendecomposition in advance using the -eigen flag. We only considered the GWAS analysis but not the data loading time, as reported by FaST-LMM.

**Code Availability**. FEATHER is available at https://github.com/cozygene/feather.

## Data availability

The informed consents given by the KORA study participants do not cover data posting in public databases. However, data are available upon request from KORA-gen (http://www.helmholtz-muenchen.de/kora-gen). Data requests can be submitted online and are subject to approval by the Steering Committee of the Research Network for Community Medicine (for SHIP data) and the KORA Board. All other relevant data are available upon request.

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

## Acknowledgements

The authors would like to thank Jennifer Listgarten for valuable suggestions. R.S. is supported by the Colton Family Foundation. E.H. and E.R. were partially supported by National Science Foundation (NSF) grant 1705197. E.R. and R.S. were supported in part by the Israel Science Foundation (Grant 1425/13) and by the Edmond J. Safra Center for Bioinformatics at Tel-Aviv University. The KORA study was initiated and financed by the Helmholtz Zentrum München German Research Center for Environmental Health, which is funded by the German Federal Ministry of Education and Research (BMBF) and by the State of Bavaria. Furthermore, KORA research was supported within the Munich Center of Health Sciences (MC-Health), Ludwig-Maximilians-Universität, as part of LMUinnovativ.

## Author contributions

R.S. led the development of the statistical method, ran the simulations, wrote the code, analyzed the simulated and real data, and led the writing of the manuscript. M.M.-N., S.K., C.G. and M.W. collected the K.O.R.A. dataset. R.S., E.F., O.W., E.R., S.R. and E.H. contributed to method development and validation, and wrote the manuscript with input from all co-authors. E.H. supervised the project.

## Additional information

**Competing interests:** R.S. is an employee of MyHeritage Ltd. The remaining authors declare no competing interests.

