## [Peer Review file · Nature Communications]

Reviewers' comments:

Reviewer #1 (Remarks to the Author):

This paper has two main points. The first is that simple tests for non-zero heritability for methylation traits in the KORA data set are poorly calibrated compared to a permutation test (likely due to model mis-specification). The second is that the authors have developed a new approach to testing for heritability using permutation that, with a combination of two tricks, can give many orders of magnitude speed-up for the estimation of low p-values. The authors claim that the same approach can be used for more general set tests (e.g. replacing SKAT p-values), though this is not pursued in the paper.

The methodological work here is impressive. Permutation testing is an important tool and the primary SAMC advance here – not their own methodology, but carefully implemented and tested – looks hugely valuable. However, I am slightly concerned about the generality of the tool and whether the re-analysis of the KORA methylation data set yields new insights. This may be a limitation of my own understanding – so apologies if this is wrong.

My reading of the paper is that the authors fit a model with only a constant covariate (i.e. remove the mean effect). So features such as age, sex, smoking or a particular genotype are not considered – as would typically be in a linear mixed model. This is presumably because the permutation only makes sense under the exchangeability assumption, which (unless you can do block-wise permutations) doesn't apply with covariates. Therefore, showing that p-values estimated under a model without covariates are poorly calibrated (for example, because there is a local meQTL of large effect driving the described bimodality) feels rather like knocking down a straw man. Of course, there are lots of papers doing something like this – e.g. papers on heritability of quantitative traits or disease susceptibility. However, I feel those analysing meQTL data are unlikely to be interested in the straw-man null model presented here. There is a comment on L373 that makes me wonder if I've misunderstood the methylation analysis / method more generally. If so, then this needs to be much better drawn out in the paper.

My specific suggestions would be:

1. To analyse traits where a like-for-like estimation and testing of heritability can be performed. E.g. height, BMI (within a sex). Or at least, re-analyse data in the same way as the original authors and show that different analysis methods lead to different conclusions.
2. Explore the bimodality (or trimodality) of methylation data as potentially being caused by covariates (sex, genotype come to mind) and estimate/test residual heritability.
3. Better describe whether/how covariates can be handled in the procedures. If the answer is 'not easily', then I'm a bit uncertain about the value of the approach, particularly if covariates can explain most of the false positive sites described in the methylation data.
4. Demonstrate a real advantage of using the current approach compared to SKAT p-values in rare variant testing. Or remove the claim about rare variant testing from the paper. I think it is important to substantiate / quantify value where claimed.

Beyond these comments, the paper is well written and clear. Some of the figures could be supplemental (Fig 2, Fig 4 – or combine Fig 4 & 5) and some of the tables I would rather see displayed graphically (e.g. Table 3).

Minor comments:

There is a statement (L382) about requiring the likelihood function to be well behaved. Can we assume this? When would it not hold?

I can see no mention of software availability. This is essential.

Reviewer #2 (Remarks to the Author):

This manuscript proposes an efficient permutation strategy for testing SNP-based heritability of complex quantitative traits, which remains valid in situations where the underlying normality assumptions of parametric methods like GCTA are violated. The authors evaluate their approach primarily using a GWAS study of

~430K methylation sites from the KORA study. The paper possesses some attractive features. The proposed permutation method, which only requires the derivative of the likelihood of permuted phenotypes (thereby bypassing the computationally laborious task of parameter estimation), is quite innovative and clever. Use of the SAMC algorithm to estimate significance of a heritability estimate using a smaller proportion of permutations than what is typically required is appealing as well.

Nevertheless, the authors are applying this innovative procedure to address an issue that may not be as significant or widespread as the authors imply throughout the manuscript. The manuscript does an insufficient job describing the features of the methylation sites in KORA that result in widely disparate p-values between permutation- and parametric-based methods, which is unfortunate. Based on some of the authors' comments though, it appears that methylation profiles that are bimodal or multimodal in nature are the ones that primarily cause unreliable GCTA p-values. The authors believe such multimodal phenotypes are commonly analyzed in methylation studies. However, most epigenetic studies remove such

bimodal/trimodal methylation sites prior to analysis because they usually indicate a SNP residing somewhere in the primer. In fact, algorithms exist for identifying such clustered probe signals yielding a bimodal/trimodal distribution (see Andrews et al. 2016; Epigenetics and Chromatin 9:56 for an example); such probes are then removed as part of QC. So, in practice, it's unclear whether the majority of sites in Figures 1 and 2 that show massive differences between the parametric- and permutation-based p-values would actually be analyzed in a standard GWAS study of methylation profiles after proper QC. The authors mention that they repeated their analyses excluding ~57K known multimodal methylation profiles but they don't show a corresponding figure showing the refined results (before and after normalization) and it's unclear if additional multimodal sites (that would be likely identified with additional in-house QC procedures like the method mentioned above) remain after these ~57K known sites are removed. Related question: Of the 1,667 sites in Table 1 that were identified as significant by GCTA but not by the permutation method, how many of these were among the ~57K known multimodal methylation profiles?

If a phenotype has to follow an extreme bimodal/multimodal distribution for GCTA to fail, then the practical impact of the permutation procedure is likely limited since such phenotypes will be rarely encountered and analyzed in a GWAS. If the authors could show more general situations where parametric and permutation-based p-values yield drastically different results, such findings would certainly strengthen the paper. Examples might include simulations involving truncated normal phenotypes or normal phenotypes demonstrating leptokurtosis (known to inflate type-I error of variance-component linkage tests of continuous traits; see Allison et al. 1999; AJHG 65: 531-544).

Other Comments

- The description of the SAMC algorithm on pages 13-15 and 21-22 was unintuitive and quite difficult to follow. The description didn't provide insight into how the algorithm can save computation cost. Derivation of the probability distribution on subsets (i) was confusing. Furthermore, what is the target probability density function of the MH method, what is the proposal distribution, and why is the algorithm updating the estimates with an arbitrary gain factor?

- Table 3 compared p-values between the parametric method and permutation method with SAMC for the 10 most extreme p-values. The parametric p-values varied between (10^{-8} , 10^{-51}), while the permutation p-values hover around 10^{-9} . However, this does not necessarily imply the parametric p-values are inflated. The permutation p-values could be limited by the number of permutations that one can perform in practice (even with SAMC algorithm).

- The title of the paper implies the permutation approach is applicable both for tests of heritability as well as tests of sets of arbitrary size (e.g. gene, pathway). However, for tests of gene sets, it's not clear the proposed permutation strategy will work as random shuffling of phenotypes will ignore the shared polygenic effects (excluding the gene set that is being tested) that likely exist among individuals. See Abney

(2015; Genetic Epidemiology 39: 249) for more details.

- Second paragraph of Introduction: There are several additional set-based tests that the authors should reference (Kwee et al. AJHG 2008; Tzeng and Zhang AJHG 2007; Wu et al. AJHG 2010).

- One quantitative criterion the authors might want to consider for evaluating the convergence of SAMC algorithm is Potential Scale Reduction Factor proposed by Gelman & Rubin 1992.

- The organization of the paper could be improved. The description of the KORA dataset and the analyses performed should be presented earlier. There are redundant descriptions of the heritability model in both the Results and Methods sections. Figures showing discrepancies in p-values could be put as panels for one Figure (Figure 7 and Figure 8 should be shown together in the main context). Power curves and the likelihood curve (figure 4) could likely be relegated to supplementary information. It would likely be sufficient to present only one Figure in the main text showing the performance of SAMC. Even the supplementary figures with different tuning parameters for SAMC could be selectively shown.

Reviewer #3 (Remarks to the Author):

In this manuscript, the authors proposed a fast permutation-based p-value calculation method for testing variance components with linear mixed models (LMM). The reviewer believes the topic is timely and the proposed method and software are highly useful. The reviewer is mainly concerned with the data application section and the overall presentation of the manuscript. The manuscript is written in a fairly technical way, and may not be a great fit for the broader audience of Nature Communications in its current form. It may be more suited for specialized statistical or computational biology journal. The application is relatively "niche". The reviewer believes that there could be broader applications of the proposed method but have not been fully exploited in the current manuscript.

Specifically, the reviewer has the following suggestions with no specific ordering:

1. A major innovation of the proposed method is to calculate permutation-based p-values without obtaining the specific estimate of each permutation. Instead, the proposed method only needs to evaluate the sign of the derivative of the likelihood at the observed estimate. This seems to be a quite general permutation approach rather than just tailored for testing variance components in LMM. Could the authors extend the idea and compare the performance of the idea versus other permutation-based approaches in other model forms (like linear regression, logistic regression, etc) than LMM?
2. In the introduction, the authors reviewed several non-parametric approaches in the literature, including the ones proposed for genome-wide association or sequencing studies. However, the descriptions for other competing methods are not clear enough for readers not familiar with this literature. The reviewer was not sure what the main idea and contributions are for each of those pieces of work, and why the proposed work is unique other than being specifically proposed for LMM.
3. There are too many similar figures and those can be a bit distractive. Some figures can be combined and some show similar conclusions with improved precision. A substantial amount of information is redundant and should be put into supplemental materials.
4. Tables 3 and 4 presented the performance of SAMC and effect of quantile normalization on extreme p-values for 10 selected sites. Not sure why those 10 sites were selected. The reviewer suggests the authors to show a substantially larger number of sites than the 10 selected ones, and how many of the SAMC p-values are lying within the CIs calculated from the permutation test. This does not have to be presented in the table format.
5. As stated above, the reviewer's main concern is the data application is not interesting and broad enough for general audience of Nature Communications. Also, for real data analysis, without knowing the truth, interpreting the discrepancies between permutation-based p-values versus parametric p-values can be limited. The authors did try to analyze possible underlying reasons for p-value discrepancies. The reviewer has one suggestion. The authors may use the proposed model to analyze a GWAS data using samples with encrypted relatedness. Due to the relatedness, LMM would be a good model. Analysis with linear regression would show inflated type I error rates. If the proposed p-value calculation is well calibrated, the authors should see a QQ plot of the GWAS p-values aligned on the 45 degree line.
6. Another potential very useful application of the proposed method is in calculating cis-eQTLs based on samples with encrypted relatedness. Again, a LMM would help to account for the sample relatedness, but is generally computational intense for eQTL analysis. The proposed method could be proved useful in that application as well.
7. The reviewer would also suggest the authors to carefully go over the grammar again. A few examples of minor grammar mistakes are listed below.
Page 6, line 177, 0.1% instead of %0.1.
Page 8, Figure 1 legend, "are often smaller that the exact p-values...", "than" instead of "that".
Page 21, line 411, "according to a the target distribution", delete "a".

Reviewer #1 (Remarks to the Author):

This paper has two main points. The first is that simple tests for non-zero heritability for methylation traits in the KORA data set are poorly calibrated compared to a permutation test (likely due to model mis-specification). The second is that the authors have developed a new approach to testing for heritability using permutation that, with a combination of two tricks, can give many orders of magnitude speed-up for the estimation of low p-values. The authors claim that the same approach can be used for more general set tests (e.g. replacing SKAT p-values), though this is not pursued in the paper.

The methodological work here is impressive. Permutation testing is an important tool and the primary SAMC advance here – not their own methodology, but carefully implemented and tested – looks hugely valuable. However, I am slightly concerned about the generality of the tool and whether

the re-analysis of the KORA methylation data set yields new insights. This may be a limitation of my own understanding – so apologies if this is wrong.

My reading of the paper is that the authors fit a model with only a constant covariate (i.e. remove the mean effect). So features such as age, sex, smoking or a particular genotype are not considered – as would typically be in a linear mixed model. This is presumably because the permutation only makes sense under the exchangeability assumption, which (unless you can do block-wise permutations) doesn't apply with covariates. Therefore, showing that p-values estimated under a model without covariates are poorly calibrated (for example, because there is a local meQTL of large effect driving the described bimodality) feels rather like knocking down a straw man. Of course, there are lots of papers doing something like this – e.g. papers on heritability of quantitative traits or disease susceptibility. However, I feel those analysing meQTL data are unlikely to be interested in the straw-man null model presented here. There is a comment on L373 that makes me wonder if I've misunderstood the methylation analysis / method more generally. If so, then this needs to be much better drawn out in the paper.

We thank the reviewer for recognizing the importance of our study, and for raising this important question. We agree that it is important to verify that our results are relevant in real research scenarios. To this end, we re-ran a complete analysis of the KORA dataset, this time including age, sex and smoking status as covariates, to demonstrate a realistic meQTL study, as suggested by the reviewer. The results are reported in the updated manuscript (Fig. S7); in short, the p-value discrepancies remain large and abundant.

To further demonstrate the generality of our results, we added a full cis-eQTL and heritability analysis of gene expression profiles in the GTEx dataset, using the exact same covariates and QC preprocessing as in the original GTEx analysis. This shows the discrepancy is a general phenomenon not limited to methylation studies. We note that as we do not know the biological truth in advance, we can only compare the two approaches and report the discrepancies. The full results are reported in the updated manuscript (mainly in Fig. 2); briefly, as in the case of methylation, p-value discrepancies remain.

My specific suggestions would be:

1. To analyse traits where a like-for-like estimation and testing of heritability can be performed. E.g. height, BMI (within a sex). Or at least, re-analyse data in the same way as the original authors and show that different analysis methods lead to different conclusions.

We followed the reviewer's suggestion, and in the analysis of the GTEx dataset we compared to the analysis that was done by the original authors. We note that the GTEx authors also mention in their appendix that a permutation test would be a more valid way to find significant results, thus highlighting the need for an efficient permutation test, which we provide in our manuscript.

2. Explore the bimodality (or trimodality) of methylation data as potentially being caused by covariates (sex, genotype come to mind) and estimate/test residual heritability.

We thank the reviewer for this important comment. We agree with the reviewer that it is important to assess the effect common covariates have on this phenomenon. As described above, we showed that age, sex and smoking status are not covariates whose inclusion explains away the discrepancy. We also ruled out the case of a particular genotype being the cause of discrepancy. This is done both by the existing analysis, which removed sites with a known SNP in their probe, as well as by a newly added analysis, which removed sites with a bimodal behaviour. For both of these, large discrepancies remain.

In addition, our updated manuscript contains a much more detailed analysis of the reasons underlying p-value discrepancies. In particular, it is shown that covariates or multi-modality are not

the only such reasons. We discuss a range of additional scenarios that may create these issues, in particular the phenotype or (the residuals) having a heavy-tailed distribution. Therefore, explaining multi-modality as a result of strong covariates would not explain away p-value discrepancies.

3. Better describe whether/how covariates can be handled in the procedures. If the answer is ‘not easily’, then I’m a bit uncertain about the value of the approach, particularly if covariates can explain most of the false positive sites described in the methylation data.

We thank the reviewer for this important comment. We agree with the reviewer that covariates are very important for a useful application of the method. We added a section in the Appendix discussing covariates in length. To summarize, our method, and in particular both the derivative trick and SAMC, fully support the inclusion of general covariates. The reviewer is correct in that only with a constant covariate (or none) does the exchangeability assumption hold and thus in that case we have a theoretical guarantee that the permutation test is exact. However, as we demonstrate both by a theoretical discussion and by an empirical study, in many (or most) practical settings, the permutation test with covariates remains well calibrated.

Our revised analysis of the KORA dataset with covariates, as well as the new gene expression analysis of the GTEx dataset with covariates, shows that covariates do not explain most of the false positive sites. In addition, we ruled out the case of a particular genotype being the cause of discrepancy, as described above.

We have also added support in our code for block-wise permutations, in cases where all important covariates are discrete, although we did not pursue this direction further in the manuscript.

Finally, while it is true that in some cases, we no longer have a theoretical guarantee that the permutation test is exact, we believe that in many cases, the evidence of the parametric p-value being wildly uncalibrated casts a much stronger doubt of its use than the potential theoretical lack of calibration for the permutation test for heritability. In the end, it is up to researcher to choose the test they believe to be more suitable for their study.

4. Demonstrate a real advantage of using the current approach compared to SKAT p-values in rare variant testing. Or remove the claim about rare variant testing from the paper. I think it is important to substantiate / quantify value where claimed.

We thank the reviewer for their comment. To show the value of our permutation testing approach in set-testing as well, our new cis-eQTL analysis of the GTEx dataset shows that also in the set-testing scenario, large p-value discrepancies are abundant. We note that we compared permutation p-values to those obtained by the LRT, and not SKAT. However, we have strong reasons to believe the SKAT p-values will be very similar: First, Fig. S12 shows large discrepancies with SKAT as well; Second, Fig. 4 in the Supplemental Material in *Schweiger et al., 2017 Genetics* shows that SKAT and LRT have comparable power for simulated phenotypes.

Beyond these comments, the paper is well written and clear. Some of the figures could be supplemental (Fig 2, Fig 4 – or combine Fig 4 & 5) and some of the tables I would rather see displayed graphically (e.g. Table 3).

We thank the reviewer for their appreciation, and for their suggestion for further improving the manuscript’s clarity. To this end, we removed some of the more redundant paragraphs, moved several figures, text and technical details to the Appendix, and made the paper more succinct altogether. We hope that the paper is more readable as a result.

Minor comments:

There is a statement (L382) about requiring the likelihood function to be well behaved. Can we assume this? When would it not hold?

This is indeed an important issue, as the requirement of the likelihood function to be well behaved is necessary for some of the assumptions of the method to hold. In our original submission we have a discussion about this issue in Appendix B. In short, while there is no theoretical guarantee for the likelihood function to be well behaved, our extensive simulations show that this happens in practice as far as we can test it.

I can see no mention of software availability. This is essential.

The location of the software is mentioned in the abstract. We supply an accompanying software package in Python and C++, which will be released by the time of publication.

Reviewer #2 (Remarks to the Author):

This manuscript proposes an efficient permutation strategy for testing SNP-based heritability of complex quantitative traits, which remains valid in situations where the underlying normality assumptions of parametric methods like GCTA are violated. The authors evaluate their approach primarily using a GWAS study of ~430K methylation sites from the KORA study. The paper possesses some attractive features. The proposed permutation method, which only requires the derivative of the likelihood of permuted phenotypes (thereby bypassing the computationally laborious task of parameter estimation), is quite innovative and clever. Use of the SAMC algorithm to estimate significance of a heritability estimate using a smaller proportion of permutations than what is typically required is appealing as well.

We thank the reviewer for recognizing the novelty and appeal of our work.

Nevertheless, the authors are applying this innovative procedure to address an issue that may not be as significant or widespread as the authors imply throughout the manuscript. The manuscript does an insufficient job describing the features of the methylation sites in KORA that result in widely disparate p-values between permutation- and parametric-based methods, which is unfortunate. Based on some

of the authors' comments though, it appears that methylation profiles that are bimodal or multimodal in nature are the ones that primarily cause unreliable GCTA p-values. The authors believe such multimodal phenotypes are commonly analyzed in methylation studies. However, most epigenetic studies remove such bimodal/trimodal methylation sites prior to analysis because they usually indicate a SNP residing somewhere in the primer. In fact, algorithms exist for identifying such clustered probe signals yielding a bimodal/trimodal distribution (see Andrews et al. 2016; Epigenetics and Chromatin 9:56 for an example); such probes are then removed as part of QC. So, in practice, it's unclear whether the majority of sites in Figures 1 and 2 that show massive differences between the parametric- and permutation-based p-values would actually be analyzed in a standard GWAS study of methylation profiles after proper QC. The authors mention that they repeated their analyses excluding ~57K known multimodal methylation profiles but they don't show a corresponding figure showing the refined results (before and after normalization) and it's unclear if additional multimodal sites (that would be likely identified with additional in-house QC procedures like the method mentioned above) remain after these ~57K known sites are removed. Related question: Of the 1,667 sites in Table 1 that were identified as significant by GCTA but not by the permutation method, how many of these were among the ~57K known multimodal methylation profiles?

If a phenotype has to follow an extreme bimodal/multimodal distribution for GCTA to fail, then the practical impact of the permutation procedure is likely limited since such phenotypes will be rarely encountered and analyzed in a GWAS. If the authors could show more general situations where parametric and permutation-based p-values yield drastically different results, such findings would certainly strengthen the paper. Examples might include simulations involving truncated normal phenotypes or normal phenotypes demonstrating leptokurtosis (known to inflate type-I error of variance component linkage tests of continuous traits; see Allison et al. 1999; AJHG 65: 531-544).

We thank the reviewer for their excellent comments and questions. We fully agree that, for the method to be widely applicable, it should apply in cases beyond that of a multimodal phenotype. To this end, we added a new section to the Appendix, with a discussion and an extensive simulation study designed to characterize when do these discrepancies occur. We have also added the Figure of the analysis excluding the ~57K known sites with SNP-containing probes. Of the 1,667 sites in Table 1 that were identified as significant by GCTA but not by the permutation method, 540 of sites with discrepancy were among these ~57K sites.

In addition, we used the method of *Andrews et al. 2016; Epigenetics and Chromatin 9:56*, suggested by the reviewer, to exclude all sites exhibiting a multimodal behaviour, and show that large discrepancies are common in the remaining sites. We concluded that discrepancies may occur also in unimodal sites with small amount of outliers.

We then performed a simulation study where phenotypes were drawn from exchangeable, yet non-normal distributions (with excess kurtosis, as suggested by the reviewer, and other distributions as well), and show that parametric p-values may be drastically uncalibrated also in a unimodal distribution. Finally, we added a gene expression analysis of the GTEx dataset, in which phenotypes were quantile-normalized, and thus were unimodal by definition. We also observed discrepancies there, adding more evidence that issues may occur with unimodality as well.

To summarize, we have shown discrepancies in the following cases:

- With only a constant covariate (KORA)
- With commonly used covariates (KORA, GTEx)
- With a small sample size and heritability estimates of 100% (GTEx)
- With quantile normalization, either without covariates (KORA) or with (GTEx)
- Compared with SKAT p-values (KORA)
- In a set-testing cis-eQTL setting (GTEx)
- When removing ~57K known sites with artifacts (KORA)
- When removing ~3.2K sites found to be multimodal (KORA)
- With phenotypes coming from non-normal distribution (simulation study)

We believe this extensive study proves the widespread possibility of p-value discrepancies across datasets and settings.

Other Comments

-The description of the SAMC algorithm on pages 13-15 and 21-22 was unintuitive and quite difficult to follow. The description didn't provide insight into how the algorithm can save computation cost. Derivation of the probability distribution on subsets $\psi(i)$ was confusing. Furthermore, what is the target probability density function of the MH method, what is the proposal distribution, and why is the algorithm updating the estimates with an arbitrary gain factor?

We thank the reviewer for their suggestion for improving manuscript clarity. We have added clarifications in the manuscript about the points raised by the reviewer. However, as we use the SAMC algorithm without modifications, and due to its inherent complexity, we do not attempt to provide a complete intuitive justification for it, as this would extend the manuscript considerably. To

clarify this, and to make the paper more succinct and less technical, we moved much of its description to the Appendix.

- Table 3 compared p-values between the parametric method and permutation method with SAMC for the 10 most extreme p-values. The parametric p-values varied between (10^{-8} , 10^{-51}), while the permutation p-values hover around 10^{-9} . However, this does not necessarily imply the parametric p-values are inflated. The permutation p-values could be limited by the number of permutations that one can perform in practice (even with SAMC algorithm).

We thank the reviewer for pointing out this delicate point. To clarify, due to the extremely computationally heavy nature of performing a permutation test without SAMC, we were limited to 10^9 permutation per site. The 10 sites in Table 3 were chosen to have at least $1/10^9$ permutations, since sites with a permutation p-value of $0/10^9$ have a very large CI, and have little value in validating the estimates of SAMC. We have been able to estimate p-value smaller than 10^{-9} with SAMC.

-The title of the paper implies the permutation approach is applicable both for tests of heritability as well as tests of sets of arbitrary size (e.g. gene, pathway). However, for tests of gene sets, its not clear the proposed permutation strategy will work as random shuffling of phenotypes will ignore the shared polygenic effects (excluding the gene set that is being tested) that likely exist among individuals. See Abney (2015; Genetic Epidemiology 39: 249) for more details.

This is an important point indeed and we thank the reviewer for raising this point. We discuss this issue in depth in the new Appendix section dedicated to covariates. In particular, the Abney 2015 paper deals with a different setting than in this paper, where the tested covariate is a single marker, and it is permuted relative to the phenotype, to other covariates, and to a non-trivial kinship matrix. With gene-sets, the tested covariates are used to construct the kinship matrix, relative to which the phenotype and covariates are permuted. To include shared polygenic effects in our method, one could include the first principal components as covariates, as performed in the GTEx cis-eQTL study added to manuscript. An alternative approach would be to extend the method to multiple variance components (i.e., multiple non-trivial kinship matrices). However, this is outside the scope of our paper and is a subject of future work, and we discuss this briefly in the Discussion.

-Second paragraph of Introduction: There are several additional set-based tests that the authors should reference (Kwee et al. AJHG 2008; Tzeng and Zhang AJHG 2007; Wu et al. AJHG 2010).

We thank the reviewer for suggesting these valuable references. We added these in the appropriate place in the introduction.

-One quantitative criterion the authors might want to consider for evaluating the convergence of SAMC algorithm is Potential Scale Reduction Factor proposed by Gelman & Rubin 1992.

We thank the reviewer for this suggestion. We currently limited our study of additional criteria of convergence, but we added this method in the appropriate discussion in the manuscript, for the reference of potential users of our method.

- The organization of the paper could be improved. The description of the KORA dataset and the analyses performed should be presented earlier. There are redundant descriptions of the heritability model in both the Results and Methods sections. Figures showing discrepancies in p-values could be put as panels for one Figure (Figure 7 and Figure 8 should be shown together in the main context). Power curves and the likelihood curve (figure 4) could likely be relegated to supplementary information. It would likely be sufficient to present only one Figure in the main text showing the performance of SAMC. Even the supplementary figures with different tuning parameters for SAMC could be selectively shown.

We thank the reviewer for their suggestions of improving the manuscript. We made an effort to make the paper more succinct, removing much of the material and figures to the Appendix. We hope that its readability is now improved.

Reviewer #3 (Remarks to the Author):

In this manuscript, the authors proposed a fast permutation-based p-value calculation method for testing variance components with linear mixed models (LMM). The reviewer believes the topic is timely and the proposed method and software are highly useful. The reviewer is mainly concerned with the data application section and the overall presentation of the manuscript. The manuscript is written in a fairly technical way, and may not be a great fit for the broader audience of Nature Communications in its current form. It may be more suited for specialized statistical or computational biology journal. The application is relatively “niche”. The reviewer believes that there could be boarder applications of the proposed method but have not been fully exploited in the current manuscript.

Specifically, the reviewer has the following suggestions with no specific ordering:

1. A major innovation of the proposed method is to calculate permutation-based p-values without obtaining the specific estimate of each permutation. Instead, the proposed method only needs to evaluate the sign of the derivative of the likelihood at the observed estimate. This seems to be a quite general permutation approach rather than just tailored for testing variance components in LMM. Could the authors extend the idea and compare the performance of the idea versus other permutation-based approaches in other model forms (like linear regression, logistic regression, etc) than LMM?

We thank the reviewer for recognizing the novelty of our approach. Unfortunately, this approach works well only for univariate functions (e.g. a linear or logistic regression problem with a single explaining covariate). For the case of multiple variance components, we have been able to show specific cases where the log-likelihood function surface is unimodal, but where at some values the gradient component corresponding to the tested variance component points away from the maximum. Since heritability testing is a scenario where a univariate function is highly useful, and since the extension of our method to multiple variant components requires more research, we feel that extending our work to this case is beyond the scope of the current work.

2. In the introduction, the authors reviewed several non-parametric approaches in the literature, including the ones proposed for genome-wide association or sequencing studies. However, the descriptions for other competing methods are not clear enough for readers not familiar with this literature. The reviewer was not sure what the main idea and contributions are for each of those pieces of work, and why the proposed work is unique other than being specifically proposed for LMM.

We thank the reviewer for their suggestion for improving manuscript clarity. We revised the relevant part and updated the introduction accordingly. We hope that the literature review is clearer now.

3. There are too many similar figures and those can be a bit distractive. Some figures can be combined and some show similar conclusions with improved precision. A substantial amount of information is redundant and should be put into supplemental materials.

We moved many of the material and figures to the Appendix, and we edited and shortened the manuscript to make it clearer.

4. Tables 3 and 4 presented the performance of SAMC and effect of quantile normalization on extreme p-values for 10 selected sites. Not sure why those 10 sites were selected. The reviewer

suggests the authors to show a substantially larger number of sites than the 10 selected ones, and how many of the SAMC p-values are lying within the CIs calculated from the permutation test. This does not have to be presented in the table format.

Due to the extremely computationally heavy nature of performing a permutation test without SAMC, we were limited to 10^9 permutation per site, and only to a limited number of sites. The 10 sites in Table 3 were chosen to have at least $1/10^9$ permutations, since sites with a permutation p-value of $0/10^9$ have a very large CI, and have little value in validating the estimates of SAMC.

5. As stated above, the reviewer's main concern is the data application is not interesting and broad enough for general audience of Nature Communications. Also, for real data analysis, without knowing the truth, interpreting the discrepancies between permutation-based p-values versus parametric p-values can be limited. The authors did try to analyze possible underlying reasons for p-value discrepancies. The reviewer has one suggestion. The authors may use the proposed model to analyze a GWAS data using samples with encrypted relatedness. Due to the relatedness, LMM would be a good model. Analysis with linear regression would show inflated type I error rates. If the proposed p-value calculation is well calibrated, the authors should see a QQ plot of the GWAS p-values aligned on the 45 degree line.

Unfortunately, the reviewer's suggestion is different from the heritability testing scenario we address here. The reviewer suggests an association test, which tests the significance of the fixed effect of each SNP separately. This is distinct from testing the variance component.

To show the relevance to other applications, we added both a heritability and eQTL analysis in the context of gene expression profiles of the GTEx study. As with most other methodologies proposed in statistical genetics, the analysis on real data provides supporting evidence. However, the ground truth is typically unknown - therefore, simulations are typically used to demonstrate the validity and utility of the method. In our case, the situation is better than the typical situation, since by definition the p-values generated using permutation tests are the correct p-values when the exchangeability property holds, allowing us to firmly conclude that our method provides calibrated p-values under these conditions, and that the LRT statistic is not well calibrated.

In addition, as mentioned, we moved many technical details and figures to the Appendix to make the paper less technical and more approachable.

6. Another potential very useful application of the proposed method is in calculating cis-eQTLs based on samples with encrypted relatedness. Again, a LMM would help to account for the sample relatedness, but is generally computational intense for eQTL analysis. The proposed method could be proved useful in that application as well.

We thank the reviewer for their suggestion. Following the reviewers' suggestion, we indeed added an cis-eQTL study to our work, which we hope demonstrates the utility of the method to that setting as well. As we note in our response to reviewer #2, using an LMM for a cis-eQTL amounts to testing using multiple variance components, which is currently outside of the scope of our work. To include shared polygenic effects in our method, one could include the first principal components as covariates, as performed in the cis-eQTL study added to manuscript, which uses both PCs and PEER factors.

7. The reviewer would also suggest the authors to carefully go over the grammar again. A few examples of minor grammar mistakes are listed below.

Page 6, line 177, 0.1% instead of %0.1.

Page 8, Figure 1 legend, "are often smaller that the exact p-values...", "than" instead of "that".

Page 21, line 411, "according to a the target distribution", delete "a".

We went over the manuscript carefully and corrected grammar mistakes, as well as those pointed out by the reviewer.

Reviewers' comments:

Reviewer #1 (Remarks to the Author):

The revised manuscript has addressed the issues I raised in my earlier review. It is an excellent piece of work and I hope will be widely used.

Reviewer #2 (Remarks to the Author):

This revision is substantially improved over the initial manuscript and provides compelling evidence that GCTA or other set-based methods can yield inaccurate asymptotic p-values in realistic situations. The authors are to be commended for the substantial effort required to strengthen the manuscript. Based on the revised content, we do have a few additional issues that we feel should be addressed:

- For the KORA results, the authors notes in the second full paragraph on page 6 that, in identifying features where permutation and parametric p-values yield wildly different results, 'A further inspection of phenotypes displaying such large discrepancies discovered that it is often the result of the phenotype taking disparate values for individuals who are relatively genetically distant from the rest of the sample.' Based on this statement, it seems the authors are saying that population outliers are the cause of these p-value discrepancies. This is understandable; such outliers are known to affect validity of GWAS studies in various contexts [Luca et al. AJHG 82:453-463]. That is why outliers are often detected and removed as part of standard QC. Did the authors identify such outliers (e.g. subjects with at least one PC more than 3 SD away from the mean across any significant axis of variation) and remove them prior to analysis? If they haven't, this should be done for both KORA and GTex as such QC is generally standard.
- In terms of general presentation, it would be of more practical value for readers of the paper to have the figures for the QC-based and covariate-adjusted results shown in the main text of the paper while relegating the figures for the non-QC unadjusted results to the supplement.
- By design, FEATHER needs to evaluate the derivative of the log-likelihood function based on the same assumed linear mixed model assumed by the GCTA method. Thus, the procedure doesn't appear to be completely nonparametric; rather FEATHER is performing nonparametric sampling for the distribution of the heritability estimate. Would one expect FEATHER to have appropriate type-I error if the underlying relationship between outcome and predictors are non-linear?
- Page 19: Parameters $\theta!$, ..., $\theta!!!$ are not defined at their first appearance in the main text. Shouldn't the initial uniform estimate be $1/(D + 1)$ instead of 0?

Reviewer #3 (Remarks to the Author):

None.

This revision is substantially improved over the initial manuscript and provides compelling evidence that GCTA or other set-based methods can yield inaccurate asymptotic p-values in realistic situations. The authors are to be commended for the substantial effort required to strengthen the manuscript. Based on the revised content, we do have a few additional issues that we feel should be addressed:

We thank the reviewer for their appreciation of our effort.

For the KORA results, the authors notes in the second full paragraph on page 6 that, in identifying features where permutation and parametric p-values yield wildly different results, ‘A further inspection of phenotypes displaying such large discrepancies discovered that it is often the result of the phenotype taking disparate values for individuals who are relatively genetically distant from the rest of the sample.’ Based on this statement, it seems the authors are saying that population outliers are the cause of these p-value discrepancies. This is understandable; such outliers are known to affect validity of GWAS studies in various contexts [Luca et al. AJHG 82:453-463]. That is why outliers are often detected and removed as part of standard QC. Did the authors identify such outliers (e.g. subjects with at least one PC more than 3 SD away from the mean across any significant axis of variation) and remove them prior to analysis? If they haven’t, this should be done for both KORA and GTEx as such QC is generally standard.

We thank the reviewer for this comment and agree that outliers are an important issue for discussion. The sentence noted by the reviewer was mistakenly left from the first submitted version. After the extensive revised analysis of the reasons for discrepancy, we feel it no longer summarizes our knowledge, as detailed further down in this work, and thus we chose to remove it from the updated draft. We definitely do not say that population outliers are the cause of discrepancies, as fully detailed in Appendix A.

To verify, we further examined the role of outliers in our data. The GTEx dataset has already undergone QC and did not contain any outliers. The KORA dataset did contain 2-5 outliers (depending on the cutoff). We removed all of them any re-ran a comparison between permutation and parametric p-values and verified that many large discrepancies remain. Therefore, removal of outliers does not solve the problem.

We note that unlike the parametric approach, a strength of the permutation method is that it does not explicitly require the removal of outliers, as those “outliers” no longer violate the model explicitly. To be more precise, it is possible that due to the sensitivity of the heritability estimator to outliers, the estimated heritability of a non-heritable phenotype will be much larger than the heritability estimated without those outliers, possibly appearing significant. However, this excessive heritability will be marked as insignificant by the permutation test, as this excessive heritability will be estimated in many of the permuted phenotypes.

In terms of general presentation, it would be of more practical value for readers of the paper to have the figures for the QC-based and covariate-adjusted results shown in the main text of the paper while relegating the figures for the non-QC unadjusted results to the supplement.

We replaced the figure in the main text with the two figures as suggested by the reviewer.

By design, FEATHER needs to evaluate the derivative of the log-likelihood function based on the same assumed linear mixed model assumed by the GCTA method. Thus, the procedure doesn’t appear to be completely nonparametric; rather FEATHER is performing nonparametric sampling for the distribution of the heritability estimate. Would one expect FEATHER to have appropriate type-I error if the underlying relationship between outcome and predictors are non-linear?

We thank the reviewer for raising this fine point. Indeed, as we detail in the Discussion, many alternative statistics can be used to measure the correspondence between phenotype and the tested markers. By definition, the permutation test will maintain its validity regardless of the metric or statistic used, and thus the type-I error will remain calibrated. It is true that FEATHER is performing nonparametric sampling from the distribution of the heritability estimate. However, the distribution it is sampling from is not the null distribution of the heritability estimator under the LMM, but rather the uniform distribution over all heritability estimates obtained by uniformly and randomly permuting the phenotype. This distinction gives FEATHER its advantage.

We chose of the LMM estimator as a statistic because (i) it has been shown to be the most powerful (or at least, one of the most powerful) alternatives; and (ii) because of its popularity in statistical genetics. We hope to incorporate additional metrics of heritability in future versions of the software.

Page 19: Parameters $\theta_1, \dots, \theta_D$ are not defined at their first appearance in the main text. Shouldn’t the initial uniform estimate be $1/(D + 1)$ instead of 0?

We thank the reviewer for noting this. We added a clarification in the main text. The parameters are in log scale and are specified up to a multiplicative (in log scale, additive) constant. Thus, an initialization of 0 is equivalent to a uniform estimate of $1/(D+1)$.